# Deciphering Mechanisms of Action of Sortilin/Neurotensin Receptor-3 in the Proliferation Regulation of Colorectal and Other Cancers

**DOI:** 10.3390/ijms231911888

**Published:** 2022-10-06

**Authors:** Jean Mazella

**Affiliations:** CNRS, Institut de Pharmacologie Moléculaire et Cellulaire, UMR 7275, Université Côte d’Azur, OK660 Route des Lucioles, 06560 Valbonne, France; mazella@ipmc.cnrs.fr; Tel.: +33-4-9395-7761; Fax: 33-4-9395-7708

**Keywords:** sortilin, neurotensin, neurotensin receptor-3, soluble sortilin, colorectal cancer, cell signaling, cell morphology

## Abstract

The purpose of this review is to decipher the mechanisms of the pathways leading to the complex roles of neurotensin (NTS) receptor-3, also called sortilin, and of its soluble counterpart (sSortilin/NTSR3) in a large amount of physiological and pathological functions, particularly in cancer progression and metastasis. Sortilin/NTSR3 belongs to the family of type I transmembrane proteins that can be shed to release its extracellular domain from all the cells expressing the protein. Since its discovery, extensive investigations into the role of both forms of Sortilin/NTSR3 (membrane-bound and soluble form) have demonstrated their involvement in many pathophysiological processes from cancer development to cardiovascular diseases, Alzheimer’s disease, diabetes, and major depression. This review focuses particularly on the implication of membrane-bound and soluble Sortilin/NTSR3 in colorectal cancer tissues and cells depending on its ability to be associated either to neurotrophins (NTs) or to NTS receptors, as well as to other cellular components such as integrins. At the end of the review, some hypotheses are suggested to counteract the deleterious effects of these proteins in order to develop effective anti-cancer treatments.

## 1. Introduction

The development of cancerous tumors is known to be the consequence of the overexpression of growth factors. Unfortunately, when treated with radiotherapy or chemical therapy, some tumors can metastasize as a result of the weakening of cancer cell–cell interactions in the tumor tissue, leading to the dissemination of cancer cells in the circulation [1,2]. Both mechanisms of cancer growth and metastasis are regulated by a large panel of circulating activators from several neuropeptides [3,4] to membrane-bound factors released by matrix metalloprotease (MMP)-dependent shedding [5], such as Epidermal Growth Factor Receptor (EGFR) ligands [6,7]. One of the most studied neuropeptides involved in cancer progression is neurotensin (NTS), the three known receptors of which (two G-protein coupled receptors, NTSR1 and NTSR2, and a type I receptor, NTSR3) are expressed in numerous cancers and particularly in digestive cancers [8,9,10,11]. Interestingly, NTSR3 [12], also previously identified as Sortilin [13], is shed from the plasma membrane [14], leading to the release of a soluble form of sortilin (sSortilin). However, growing evidence indicates the emerging role of membrane-bound Sortilin/NTSR3 and its soluble counterpart in cancer cell proliferation and dissemination.

The identification of Sortilin/NTSR3 by three different experimental approaches predicted the complexity of the functions of the protein. Chronologically, by using the chaperon protein RAP (receptor-associated protein) affinity column, Petersen and collaborators identified and purified a 95 kDa protein from human brain extract using the detergent CHAPS. Molecular cloning of the encoding gene showed that the protein was a type I receptor with homology to the yeast vacuolar protein sorting 10 protein (Vps10p) of the sorting proteins family [13]. On another hand, by using an affinity column made with antibodies against the glucose transporter Glut4, Kandror’s team identified a glycoprotein of 110 kDa as a major component of Glut4-containing vesicles. The molecular cloning of sortilin protein from rat adipocytes indicated a 93% identity to human sortilin [15]. Finally, at the same time, the 100 kDa protein previously identified as the NTS binding site was purified from CHAPS-solubilized mouse and human brain extracts by using an NTS affinity column. The cloning of the human 100 kDa NTS binding site identified the protein as sortilin [12].

Therefore, since its discovery, several cellular functions have been described for Sortilin/NTSR3, including the sorting of proteins to the plasma membrane [16,17] or to lysosomes [18,19]. In addition to its involvement in intracellular trafficking, Sortilin/NTSR3 also displays a receptor function for NTS [20], for a lipoprotein lipase [21], and a co-receptor function to initiate the action of NTS in pancreatic beta cells [22,23,24], as well as in the HT29 adenocarcinoma colorectal cancer cells [25,26]. Additionally, Sortilin/NTSR3 has been shown to interact with the receptor of Nerve Growth Factor (NGF), the p75 neurotrophin receptor (p75NTR), to trigger neuronal apoptosis induced by the precursor of Nerve Growth Factor (pro-NGF) [27,28] and the precursor of Brain-Derived Neurotrophic Factor (pro-BDNF) [29,30]. Recently, it was identified that Sortilin/NTSR3 acts as a receptor for the brain lipids carrier Apolipoprotein E (apoE), which confers the most important genetic risk factor for Alzheimer’s disease (AD), demonstrating the involvement of Sortilin/NTSR3 in the neuroprotective action of apoE in AD pathology [31]. Finally, the role of Sortilin/NTSR3 as a biomarker of risk in cardiovascular disorders in humans has been largely confirmed [32,33].

In the field of cancer, the initial involvement of Sortilin/NTSR3 has been observed by its implication in the NTS-induced proliferation of several cancer cell lines [9]. Subsequently, a series of works were developed to investigate the role of Sortilin/NTSR3 either as an actor or as a biomarker in the development of human cancers [34]. Briefly, the overexpression of Sortilin/NTSR3 is linked to proliferation and migration in neuroendocrine tumors [35], in breast and ovarian carcinomas [36,37,38], in gliomas [39,40,41], in thyroid cancers [42], and in chronic lymphocytic leukemias [43].

This review focuses on the role of the two predominant protein forms of sortilin (the membrane-bound and the soluble form), particularly in colorectal cancer, through their interaction with various types of membrane receptors such as NTSR1, epidermal growth factor receptor (EGFR), and tropomyosin receptor kinase B (TrkB).

## 2. The Membrane-Bound Sortilin/NTSR3

At the cellular level, from its translation to its targeting to the plasma membrane, Sortilin/NTSR3 has been described to be classically associated with the membrane in the endoplasmic reticulum, the trans-Golgi network, and finally, in the plasma membrane after transport in membrane vesicles. Note that the three independent works described above that led to the cloning of Sortilin/NTSR3 indicated that only 10% of the protein was associated with the surface membrane, whilst 90% of the protein remained intracellular [12,13,15]. Once at the cell surface, Sortilin/NTSR3 could be released into the circulation by shedding [14,44] or released in extracellular micro-vesicles termed exosomes [45].

### 2.1. The Role of Membrane Sortilin/NTSR3 in the Signaling and Trafficking of Neurotrophin Receptors

Neurotrophins (NTs) are growth factors that control a series of functions in the nervous system. The mature forms of NGF and BDNF, as well as those of NT4/5 and NT3, are involved in Trk-dependent neuronal cell survival, whereas their unmatured forms are responsible for cell death through p75NTR [46]. In fact, all the functions described above for NT receptors necessitate their association with Sortilin/NTSR3, which was well described in a previous review [47]. Focusing on colorectal cancer cells, the interaction of Sortilin/NTSR3 with either TrkB or p75NTR, both expressed in colorectal cancer cells, triggers opposite functions. On the one hand, BDNF, the secretion of which is activated by Sortilin/NTSR3 [48], induces cell proliferation and displays anti-apoptotic effects through TrkB [49]. One the other hand, exogenous pro-BDNF induced colorectal cancer cell apoptosis through Sortilin/NTSR3 as a co-receptor of p75NTR, the high-affinity receptor for pro-neurotrophins, suggesting a mechanism of Sortilin/NTSR3 action that can counterbalance cell survival [49] (Figure 1).

### 2.2. The Role of Membrane Sortilin/NTSR3 in the Signaling and Trafficking of Neurotensin Receptors

NTS and its receptors NTSR1 and Sortilin/NTSR3 are significantly overexpressed in colorectal cancer cells when compared to the surrounding normal epithelium, an observation that can potentially be used as a prognostic biomarker associated with more advanced colorectal cancer and poorer disease-free survival [50]. 

In the human colonic adenocarcinoma cell line HT29, Sortilin/NTSR3 is co-expressed with the G-protein coupled receptor NTSR1 (Figure 2). Immunoprecipitation experiments provided evidence for endogenous complex formation between these two receptors. It has also been demonstrated that the NTSR1–Sortilin/NTSR3 complex is internalized on NTS stimulation [25]. More interestingly, the interaction of Sortilin/NTSR3 with NTSR1 modulates both the NTS-induced phosphorylation of mitogen-activated protein (MAP) kinases and the phosphoinositide (PI) turnover mediated by NTSR1 [51], suggesting that Sortilin/NTSR3 may act as a co-receptor to participate in true NTS signaling. To further examine the functionality of Sortilin/NTSR3 trafficking in HT29 cells, the internalization of the Sortilin/NTSR3–NTS complex was followed from the plasma membrane to the trans-Golgi network (TGN), where NTS was bound to a lower molecular form of the receptor compared to the form found at the cell surface or on early endosomes [51]. This result suggested that the signaling and transportation functions of Sortilin/NTSR3 may be mediated through different molecular forms of the protein, a high-molecular-weight membrane form responsible for NTS endocytosis and a low-molecular-weight intracellular form responsible for the sorting of internalized NTS to the TGN. Once again, the role of Sortilin/NTSR3 in HT29 proliferation appears rather essential in the regulation of the action of NTS to modulate cancer cell proliferation.

In the same colorectal cell line, a study from Navarro et al. demonstrated that NTS-induced proliferation was dependent on the internalization of the Sortilin/NTSR3-NTS complex [52]. Inhibition of the internalization process affected NTS-induced Erk1/2 phosphorylation and cell proliferation, whereas the peptide-induced activation of phospholipase C was unaffected, indicating that the two intracellular pathways activated by NTS in HT29 cells (phospholipase C and MAP kinases) are independent. This can be explained by distinct conformational structures formed by the associated NTSR1 and Sortilin/NTSR3, leading to either G-protein activation or to the process of sequestration. This also indicates that inhibiting the trafficking of surface protein receptors could be an alternative method through which to develop anti-cancer treatments.

## 3. The Soluble Form of Sortilin/NTSR3

### 3.1. Shedding of the Cell Surface Sortilin/NTSR3 

The shedding of Sortilin/NTSR3 was not stimulated by NTS itself, but the amount of shed protein (sSortilin/NTSR3) recovered in the extracellular medium was enhanced when the internalization process was blocked by hyperosmolar sucrose suggesting an accumulation of the protein at the cell surface and also an increase in the amount of shed protein in these conditions. The shedding process of Sortilin/NTSR3 is activated in a concentration- and time-dependent manner by PMA (Phorbol 12-Myristate 13-Acetate), a protein with a molecular weight of 100 kDa, which is slightly lower than that detected in crude homogenates (110 kDa). PMA acts as an activator of MMPs via the PKC pathway in several types of cells such as neurons, microglial cells, and cancer cells [14]. In the same way, other PKC activators such as carbachol or PGE2 [53] increased the shedding of Sortilin/NTSR3 [54]. Note that other members of the Vps10p receptor family, SorLA and SorCS1-3, are also shed [44,55].

### 3.2. Binding and Internalization Properties of sSortilin/NTSR3 

The shed Vps10p proteins could display their own activities as ligands or could serve as transporters/protectors to avoid the proteolytic degradation of their ligands. Binding experiments performed on HT29 cell homogenates using ^125^I-radiolabeled proteins showed that sSortilin/NTSR3 specifically bound to HT29 membranes with an affinity of 5 nM but not to the other NTS receptors [54]. Although in numerous cancer cell systems, NTS signaling depends on EGFR activation [56,57], this is not the case in HT29 cells, since sSortilin/NTSR3 is unable to compete with EGF on the EGFR, and reciprocally, EGF is unable to compete with sSortilin/NTSR3 on its binding sites [54]. These results indicate that sSortilin/NTSR3 recognizes a specific receptor in HT29 cells that is neither sortilin nor EGFR (Figure 2).

After binding to a specific receptor, sSortilin/NTSR3 is rapidly and efficiently sequestrated at 37 °C into HT29 cells by a mechanism dependent on hyperosmolar sucrose [54]. Following its internalization, 60–70% of the sequestered protein is recovered into lysosomes and degraded. The remaining non-degraded sSortilin/NTSR3 could be sorted to recycling vesicles or to other cellular compartments to trigger unidentified functions. The intracellular fate of sSortilin/NTSR3 appears to follow the same sorting to lysosomes that the membrane-bound Sortilin/NTSR3 undergoes [58].

### 3.3. Cell Functions of sSortilin/NTSR3 in HT29 Cells

sSortilin/NTSR3 induces plasma membrane translocation of PKCα and consequently increases the intracellular concentration of calcium at low concentrations (10 nM) [54]. It was shown that the effect of sSortilin/NTSR3 on calcium concentrations can be desensitized, a mechanism frequently observed by the internalization and uncoupling of functional receptors such as G-protein coupled receptors [59] and the low-density lipoprotein lipase receptor family [60].

In HT29 cells, sSortilin/NTSR3 rapidly and transiently activates Akt phosphorylation through the upstream phosphorylation of the complex focal adhesion kinase FAK-Src [54]. The activation of the phosphatidylinositol 3-kinase (PI3 kinase) pathway is an important step to induce calcium release from the intracellular stores (for a review, see [61]), a pathway involved in the development of colorectal cancers [62]. It is important to note that the activation of the FAK pathway is involved in survival mechanisms, and especially in a variety of distinct cancer cell development and metastasis processes [63,64].

### 3.4. Morphological Changes of HT29 Cells Induced by sSortilin/NTSR3

The activation of the focal adhesion kinase (FAK) pathway is known to be correlated with numerous cellular processes such as cell spreading, adhesion, migration, and survival [65]. In HT29 cells, the shape and the morphology on sSortilin/NTSR3 incubation were investigated to determine the role of the protein in the regulation of cancer cell detachment [66].

The geometric distribution (polygon classes) of cells, assessed using labeling with fluorescent anti-E-cadherin antibodies, illustrates that resting confluent HT29 cells presented a geometric distribution corresponding to 46% hexagons, a distribution in agreement with several other resting cells [67,68]. Interestingly, sSortilin/NTSR3-treated HT29 cells displayed a significant reduction (to 30%) in the proportion of hexagons in favor of pentagons, as well as an increase in the cell surface [66].

The modifications by sSortilin/NTSR3 of the actin cytoskeleton and the cell shape, as well as its ability to activate FAK, were likely linked to the cell–matrix contact weakening, which can lead to cell migration. However, HT29 cells are non-migrating cells [69]; therefore, the role of sSortilin/NTSR3 could correspond to involvement in the first step of a mechanism responsible for cell detachment.

Could the reorganization of the cell shape by sSortilin/NTSR3 be in agreement with the modifications of the architecture of ultra-structural components such as desmosomes and intermediate filaments? [66]. To answer this question, the number and structure of desmosomes have been analyzed. Desmosomes are formed by plaque densities and bundles of intermediate filaments. These structures are involved in cell–cell adhesion by connecting the proteins forming plaque densities to the interfilaments’ cytoskeleton. The desmosomes are important to ensure tissue integrity and to maintain homeostasis [70]. In fact, sSortilin/NTSR3 decreases the average number of desmosomes per cell and modifies the architecture of desmosomes, thus weakening the cell–cell and cell–matrix interactions (Figure 2). The disorganization of desmosomes may contribute to the weakening of the cell barrier, which can allow for the crossing of growth factors leading to tissue dysfunction, particularly in the development or progression of human epithelial cancer cells (for reviews see [71,72]).

The marked changes observed in the sSortilin/NTSR3-treated HT29 cell morphology were correlated with the decreased expression of E-cadherin and a series of integrin family members, proteins implicated in cell–cell junctions or cell adhesion [66,73]. A decrease in or loss of integrin couples has already been described in colonic epithelial cells [74,75] in association with a poor prognosis.

Cell detachment from the plates has previously been observed in resting colonic cancer cells including HCT116, HT29, and SW620 cell lines. The action of sSortilin/NTSR3 in the weakening of cell–cell contact and cell–matrix interactions may be part of a mechanism responsible for the initial steps leading to cancer cell detachment and diffusion from primary tumors to healthy non-tumoral cells, thus facilitating metastasis (Figure 2).

## 4. Another Crucial Function of Sortilin/NTSR3: Possible Role in the Field of Cancer

### Involvement of Sortilin/NTSR3 in the Membrane Expression of TREK-1OK

A previous study demonstrated the interaction between the two proteins sortilin and TREK-1, in which mice with deletions of the sortilin (*sort1*) or TREK-1 (*kcnk2*) genes displayed a similar phenotype of resistance to depressive-like behavior during resignation tests such as the forced swimming test (FST) and the tail suspension test (TST) [76]. TREK-1 belongs to the family of two-pore-domain potassium channels, which play important roles in neuroprotectioTablen, pain, analgesia, and depression [77,78,79]. As one of the first functions identified for Sortilin/NTSR3 was to address numerous proteins from the intracellular compartments to the plasma membrane or lysosomes [17,80], it was crucial to determine whether sortilin and TREK-1 were associated, and if they were, how sortilin was involved in the sorting of TREK-1. This hypothesis was firstly confirmed by demonstrating that the TREK-1 channel expression at the plasma membrane of COS-7 cells was strongly enhanced by the co-expression of Sortilin/NTSR3 [76], and secondly, by observing that the brain of *sort1−/−* mice had an altered TREK-1 function due to a dramatically lower expression of the channel at the plasma membrane of neurons [81]. Therefore, the regulation of the functional expression of TREK-1 could be of importance in a series of human cancers, including prostate [82,83] and endometrial [84] cancers, in which the overexpression of the potassium channel appears to be responsible for tumor development. The ability of spadin, a shorter analog of the pro-peptide (PE) released from the maturation of Sortilin/NTSR3 to block the activity of TREK1, indicates that it could possibly be used as a tool to decrease the proliferation of cancer cells.

## 5. Conclusions

The initial multiplicity of functions described for Sortilin/NTSR3 has been enhanced by the protein kinase C-dependent shedding of the membrane-bound protein, leading to a soluble extracellular form for which additional actions have been observed. The regulation of both forms involves complex mechanisms. From the various functions of Sortilin/NTSR3 in numerous cell types and tissues, the implications of membrane-bound and soluble Sortilin/NTSR3 in CRC cells are summarized in Table 1.

Focusing on cancers, and particularly on colorectal cancers, most studies on the role of Sortilin/NTSR3 have been performed on in vitro models (human colorectal cancer cell lines). A recent study carried out on both cell lines and primary cultures from patients demonstrated that the overexpression of Sortilin/NTSR3 was associated with 5-fluorouracil (5-FU) resistance and a poor prognosis in colorectal cancer [85].

At the plasma membrane level, the association of Sortilin/NTSR3 with the neurotrophin receptors TrkA/B/C induced cell proliferation, whereas its association with the neurotrophin receptor p75NTR triggered cell death (Figure 1). In addition, p75NTR can dimerize with other Trk receptors, as well as with NTS receptors 1 and 2 (NTSR1-2) [86]. The observation that p75NTR can undergo ectodomain shedding by γ-secretase and TNFα-convertases, a cleavage that abolishes ligand-induced signaling and produces an active intracellular fragment, increases the complexity of the role of these proteins either in cell survival or in cell death [87].

Although not demonstrated in all cases, the role of the complex Sortilin/NTSR3–NTSR1 in NTS-induced cancer cell proliferation was described to be dependent on the internalization process, at least in the HT29 cell line [52], thus increasing the complexity of the mechanisms of action that regulate cancer cell growth. Further investigations regarding the importance of the internalization process in cancer cell proliferation should assist in the identification of new molecular target(s) to counteract cancer development.

The cleavage of the membrane Sortilin/NTSR3 by a mechanism dependent on the activation of protein kinase C leads to the release of the luminal part of Sortilin/NTSR3. Then, sSortilin/NTSR3 can bind to specific binding sites to participate in several intracellular signaling pathways including the activation of PKCα, an autoregulation process of the expression of the soluble protein in the HT29 cell line. The effective concentrations of sSortilin/NTSR3 that induce PKCα activation and calcium increase are around 10 nM, concentrations that are in accordance with both circulating serum levels of sSortilin/NTSR3 determined in several previous works and with the affinity of sSortilin/NTSR3 to its unidentified receptor [54].

In order to control the expression of Sortilin/NTSR3 and its soluble counterpart, regarding their positive or negative behavior in the development of numerous pathologies, the use of specific inhibitors of the proteins, their unknown receptors, and/or their associated co-receptors would ameliorate treatments, particularly in cancers. However, it is important to avoid targeting sortilin non-selectively due to its physiological expression and function in numerous crucial tissues. In the present case of colorectal cancers, the use of a cytotoxic agent conjugated to NTS, for example, can provide an efficient treatment against tumor development, as shown in breast cancer [88]. Another possibility could be the use of the natural NTS antagonist pro-peptide (PE) released from the maturation of the sortilin precursor, which has been shown to counteract the activation of microglial cells by NTS [20].

The complex formed between Sortilin/NTSR3 and integrin(s) as a receptor for sSortilin/NTSR3 may be responsible for the activation of FAK, as observed in HT29 cells, since integrins are described as being able to stimulate intracellular kinases including FAK-Src (for a review, see [89]). In agreement with this hypothesis, this complex can ensure strong links between cells, which can be weakened by sSortilin/NTSR3 competition with the membrane sortilin, a process leading to the dissociation and dissemination of cancer cells.

Finally, since sSortilin/NTSR3 exhibits important roles in the development of metastasis, and given that its serum levels are deleterious in other important diseases such as cardiovascular disease and depression [90,91,92,93,94,95], regulation of the soluble protein formation appears to be possible by targeting the activity of MMPs. Indeed, the use of BB3103, the ADAM10 (A Desintegrin And Metalloprotease) inhibitor, was shown to block the formation of sSortilin/NTSR3 by HT29 cells [14]. Although clinical trials targeting MMPs have been canceled in phase I or in phase III for numerous inhibitors, increasing the selectivity for specific MMPs could offer a useful objective for the further development of selective treatments for each type of cancer (for a review see, [96]).

## Figures and Tables

**Figure 1 ijms-23-11888-f001:**
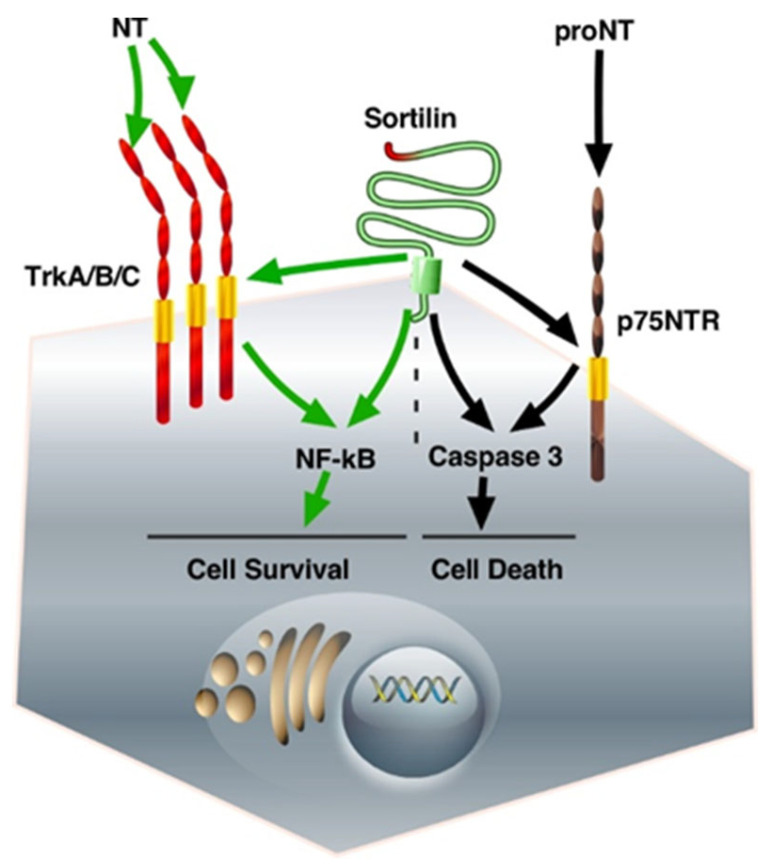
The fate of neurotrophin receptors/Sortilin interactions. The binding of proNT to the complex Sortilin/p75NTR triggers intracellular pathways that activate caspase 3, leading to cell death. On the other hand, the binding of matured NT to Trk receptors is responsible for NF-kB activation, which induces cell survival. The abbreviations used are: Trk, Tropomyosin Receptor Kinase; NT, neurotrophin; proNT, neurotrophin precursor; p75NTR, p75 neurotrophin receptor; NF-kB, nuclear factor kappa-light chain enhancer of activated B cells.

**Figure 2 ijms-23-11888-f002:**
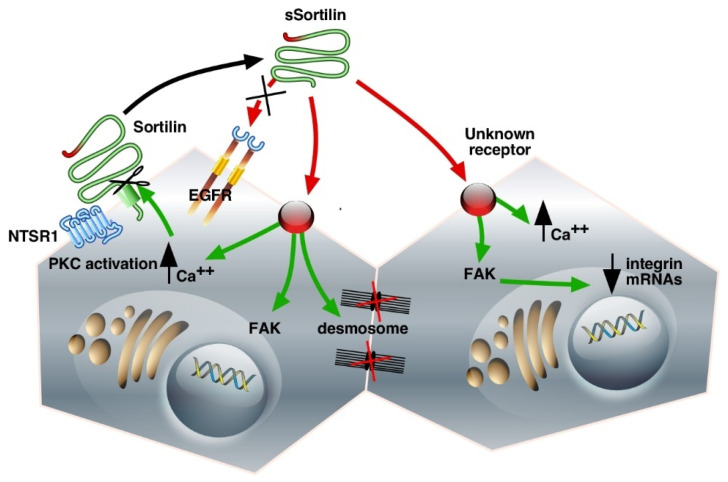
Schematic representation of Sortilin/NTSR3 shedding and signaling cascades in the HT29 cell line. The interaction of the membrane-bound Sortilin/NTSR3 with NTSR1 leads to the modulation of NTS-induced PKC activation. The shedding of Sortilin/NTSR3 releases sSortilin, which can interact with unknown receptors to increase intracellular Ca++, activate FAK, decrease integrin mRNAs, and lead to desmosome disruption. Green, intracellular signaling pathways; red, extracellular interactions. The abbreviations used are: NTSR1, neurotensin receptor-1; Sortilin/NTSR3, neurotensin receptor-3/sortilin; sSortilin/NTSR3, soluble neurotensin receptor-3/sortilin; EGFR, Epidermal Growth Factor Receptor; PKC, Protein Kinase C; FAK, Focal Adhesion Kinase.

**Table 1 ijms-23-11888-t001:** Implication of membrane-bound and soluble Sortilin/NTSR3 in CRC cells.

Membrane-Bound Sortilin as a Co-Receptor		
Ligand and receptor	Function	Pathways
NTS, NTSR1 BDNF, TrkB Pro-BDNF, p75NTR	Cell proliferation Cell proliferation, anti-apoptotic Cell apoptosis	PKC, ERK1/2, PI3K/Akt PI3K/Akt
**Soluble Sortilin** **as a Ligand**		
Receptor	Function	Pathways
Unknown Unknown Unknown, EGFR- independent	Cell-cell disruption Cell morphological changes Cytoskeleton redistribution Cell proliferation	FAK/Src, PI3K/Akt Integrins expression changes ERK1/2, PKCα

CRC colorectal cancer, NTS neurotensin, NTSR neurotensin receptor, EGFR epidermal growth factor receptor, BDNF brain-derived neurotrophic factor, pro-BDNF precursor of BDNF, Trk tropomyosin receptor kinase, p75NTR p75 neurotrophin receptor, PKC protein kinase C, ERK extracellular signal-regulated kinase, PI3K phosphatidylinositol 3-kinase, FAK, focal adhesion kinase.

## Data Availability

Not applicable.

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
