# Peer review of "Deciphering Mechanisms of Action of Sortilin/Neurotensin Receptor-3 in the Proliferation Regulation of Colorectal and Other Cancers"

_ijms, 2022, doi:10.3390/ijms231911888_

Round 1

Reviewer 1 Report

Dear authors, 

The article titled as "The emerging role of sortilin/neurotensin receptor-3 in colorectal cancer " is intended to elucidate the roles of sortilin in colorectal cancer. The article summarized several articles to show the role in colorectal cancer. However, there are several points needed to be addressed to strengthen this paper. 

  1. Based on the title, the authors should describe more evidence of sortilin regulating colorectal cancer. However, in this article, few study described.
  2. To make it easily readable, the authors should have more pictures to show its pathway as membrane-bound or secretory sortilin. (https://doi.org/10.1038/s41419-020-03245-8). More figures would assist the readers to know and understand how important it is. 
  3. A table to explain the roles of  membrane-bound or secretory sortilin in cancer or colorectal cancer
  4. Usually, in a figure, the abbreviation should be kept in the final part but not A. ... abb. B. ....abb. 
  5. In part 4, nothing related to cancer, please delete it and find somethings worthy. 

Author Response

Reviewer 1

  1. Based on the title, the authors should describe more evidence of sortilin regulating colorectal cancer. However, in this article, few study described.

Answer to reviewer

The title of the review has been changed to cover the content of the manuscript.

  1. To make it easily readable, the authors should have more pictures to show its pathway as membrane-bound or secretory sortilin. (https://doi.org/10.1038/s41419-020-03245-8). More figures would assist the readers to know and understand how important it is. 

Answer to reviewer

As suggested, the initial figure 1 has been separated into two Figures describing in one hand the interaction of sortilin with neurotrophin receptors (Figure 1) and on the other hand, the interactions of sortilin with neurotensin receptors (Figure 2).

  1. A table to explain the roles of  membrane-bound or secretory sortilin in cancer or colorectal cancer

Answer to reviewer

A table describing the roles of membrane-bound and soluble sortilin in colorectal cancers has been added.

  1. Usually, in a figure, the abbreviation should be kept in the final part but not A. ... abb. B. ....abb. 

Answer to reviewer

????

  1. In part 4, nothing related to cancer, please delete it and find somethings worthy. 

Answer to reviewer

OK, the second part of the chapter 4 hs been deleted

Reviewer 2 Report

The aim of this review was “to decipher the mechanisms of pathways leading to the complex roles of the Neurotensin (NTS) receptor-3, also called sortilin, and of its soluble counterpart (sSortilin/NTSR3) in a large amount of physiological and pathological functions particularly in cancer progression and metastasis. The review focuses on the implication of membrane and sSortilin/NTSR3 in colorectal cancerous tissues and cells depending on their ability to be associated either to Neurotropins (NT) or to NTS receptors and also to other cellular components like integrins”.

The subject matter of the paper is certainly interesting, but too brief in the context of the stated plans for describing the role of sortilins in colorectal cancer (CRC). The title of the paper is inadequate to the content of the overly laconic summary of the work on the role of sortilins in colorectal cancer. The term "colorectal cancer" is used 12 times throughout the paper (including references). Only few papers dealing directly with colorectal cancer (including mostly in vitro models) are cited in the review (nos. 26, 49, 50, 51, 52).

Thus, the title of the paper needs to be changed or the text of the paper should emphasize more the role of sortlin in CRC.

Other comments:

  1. The lack of line numbering is a difficulty in reading the paper and adding comments.
  2. Figure 1 is inappropriately placed, especially since there is no description of sortilin action and function in cancer in its legend. Figure 1B is quoted late, only on page 5. The reader has not yet been introduced to the subject to understand this figure already on page 3. In my opinion, the figure should be placed somewhere in section 3.2 (where it is once again quoted) and should be supplemented with more details.
  3. in subsection 4.1, I do not understand the sentence: "this chapter aims to make the link between the role of.... The cited paper is also not about CRC. So how is it about CRC?
  4. the last subchapter, Conclusions, needs to be rewritten. Overall to me it is incomprehensible. It contains a lot of results of work unrelated to the role of sortilin in CRC. Please summarize only the studies related to CRC, or change the overall goals of the paper to a more general action of sortilins in (various) cancer or disease biology.
  5. please check terms of abbreviations, used for the first time in the text and standardize them, first the whole name then in brackets abbreviation (e.g. PMA), TREK-1.
  6. minor language errors (bad spelling): "identidy", "TrKB" "discribed", "pronostic"; in ref. list - "iScience".
  7. the table of references should also be standardized, according to the requirements of the journal.

Author Response

Reviewer 2

The subject matter of the paper is certainly interesting, but too brief in the context of the stated plans for describing the role of sortilins in colorectal cancer (CRC). The title of the paper is inadequate to the content of the overly laconic summary of the work on the role of sortilins in colorectal cancer. The term "colorectal cancer" is used 12 times throughout the paper (including references). Only few papers dealing directly with colorectal cancer (including mostly in vitro models) are cited in the review (nos. 26, 49, 50, 51, 52).

Answer to reviewer

The title of the review has been changed to be in adequation with the role of sortilin in colorectal and other cancers. However, in addition to the references underlined by the reviewer, many other papers dealing with colorectal cancers have been cited in the first version (refs 8, 14, 25, 47, 54, 66).

Other comments:

  1. The lack of line numbering is a difficulty in reading the paper and adding comments.

Answer to reviewer

The absence of line numbering and the difficulty to understand the Figure legend are the consequence of a bad conversion of the pdf file I sent in which both the line numbering and the figure legend were clearly presented.

  1. Figure 1 is inappropriately placed, especially since there is no description of sortilin action and function in cancer in its legend. Figure 1B is quoted late, only on page 5. The reader has not yet been introduced to the subject to understand this figure already on page 3. In my opinion, the figure should be placed somewhere in section 3.2 (where it is once again quoted) and should be supplemented with more details.

Answer to reviewer

Figure 1 has been changed in two figures to a better understanding.

  1. in subsection 4.1, I do not understand the sentence: "this chapter aims to make the link between the role of.... The cited paper is also not about CRC. So how is it about CRC?

Answer to reviewer

This chapter has been deleted in the new version

  1. the last subchapter, Conclusions, needs to be rewritten. Overall to me it is incomprehensible. It contains a lot of results of work unrelated to the role of sortilin in CRC. Please summarize only the studies related to CRC, or change the overall goals of the paper to a more general action of sortilins in (various) cancer or disease biology.

Answer to reviewer

The chapter Conclusions has been modified and more focused in relation to CRC.

  1. please check terms of abbreviations, used for the first time in the text and standardize them, first the whole name then in brackets abbreviation (e.g. PMA), TREK-1.

Answer to reviewer

OK, this was corrected

  1. minor language errors (bad spelling): "identidy", "TrKB" "discribed", "pronostic"; in ref. list - "iScience".

Answer to reviewer

OK, this was corrected

  1. the table of references should also be standardized, according to the requirements of the journal.

Answer to reviewer

OK, this was corrected